# Hypotension Prediction Index Software to Prevent Intraoperative Hypotension during Major Non-Cardiac Surgery: Protocol for a European Multicenter Prospective Observational Registry (EU-HYPROTECT)

**DOI:** 10.3390/jcm11195585

**Published:** 2022-09-23

**Authors:** Manuel Ignacio Monge García, Daniel García-López, Étienne Gayat, Michael Sander, Peter Bramlage, Elisabetta Cerutti, Simon James Davies, Abele Donati, Gaetano Draisci, Ulrich H. Frey, Eric Noll, Javier Ripollés-Melchor, Hinnerk Wulf, Bernd Saugel

**Affiliations:** 1Medical Affairs Department, Critical Care Europe, Edwards Lifesciences, 1260 Nyon, Switzerland; 2Department of Anaesthesiology and Reanimation, University Hospital Marqués de Valdecilla, 39008 Santander, Spain; 3Université Paris Cité, INSERM, 75006 Paris, France; 4Department of Anesthesia and Critical Care Medicine, Hôpital Lariboisière, 75475 Paris, France; 5Department of Anaesthesiology, Intensive Care Medicine and Pain Medicine, University Hospital Giessen, Justus-Liebig University Giessen, 35392 Giessen, Germany; 6Institute for Pharmacology and Preventive Medicine, 49661 Cloppenburg, Germany; 7Department of Anesthesia, Transplant and Surgical Intensive Care, Azienda Ospedaliero Universitaria Ancona, 60126 Ancona, Italy; 8York and Scarborough Teaching Hospitals NHS Foundation Trust, York YO31 8HE, UK; 9Centre for Population and Health Studies, Hull York Medical School, York HU6 7RU, UK; 10Department of Biomedical Sciences and Public Health, Università Politecnica delle Marche, 60126 Ancona, Italy; 11Unit of Obstetric and Gynecologic Anesthesia, IRCCS Fondazione Policlinico Universitario Agostino Gemelli, 00168 Rome, Italy; 12Department of Anesthesiology, Intensive Care, Pain and Palliative Care, Marien Hospital Herne, Ruhr-University Bochum, 44801 Bochum, Germany; 13Department of Anesthesiology and Intensive Care, Les Hôpitaux Universitaires de Strasbourg, 67098 Strasbourg, France; 14Anesthesia and Critical Care Department, Hospital Universitario Infanta Leonor, 28031 Madrid, Spain; 15Department of Anaesthesiology—Intensive Care, University Hospital Marburg, 35043 Marburg, Germany; 16Department of Anesthesiology, Center of Anesthesiology and Intensive Care Medicine, University Medical Center Hamburg-Eppendorf, 20251 Hamburg, Germany; 17Outcomes Research Consortium, Cleveland, OH 44195, USA

**Keywords:** artificial intelligence, blood pressure, hemodynamic instability, advanced hemodynamic monitoring, machine learning, postoperative complications

## Abstract

Background: Intraoperative hypotension is common in patients having non-cardiac surgery and associated with postoperative acute myocardial injury, acute kidney injury, and mortality. Avoiding intraoperative hypotension is a complex task for anesthesiologists. Using artificial intelligence to predict hypotension from clinical and hemodynamic data is an innovative and intriguing approach. The Acumen^TM^ Hypotension Prediction Index (HPI) software (Edwards Lifesciences; Irvine, CA, USA) was developed using artificial intelligence—specifically machine learning—and predicts hypotension from blood pressure waveform features. We aimed to describe the incidence, duration, severity, and causes of intraoperative hypotension when using HPI monitoring in patients having elective major non-cardiac surgery. Methods: We built up a European, multicenter, prospective, observational registry including at least 700 evaluable patients from five European countries. The registry includes consenting adults (≥18 years) who were scheduled for elective major non-cardiac surgery under general anesthesia that was expected to last at least 120 min and in whom arterial catheter placement and HPI monitoring was planned. The major objectives are to quantify and characterize intraoperative hypotension (defined as a mean arterial pressure [MAP] < 65 mmHg) when using HPI monitoring. This includes the time-weighted average (TWA) MAP < 65 mmHg, area under a MAP of 65 mmHg, the number of episodes of a MAP < 65 mmHg, the proportion of patients with at least one episode (1 min or more) of a MAP < 65 mmHg, and the absolute maximum decrease below a MAP of 65 mmHg. In addition, we will assess causes of intraoperative hypotension and investigate associations between intraoperative hypotension and postoperative outcomes. Discussion: There are only sparse data on the effect of using HPI monitoring on intraoperative hypotension in patients having elective major non-cardiac surgery. Therefore, we built up a European, multicenter, prospective, observational registry to describe the incidence, duration, severity, and causes of intraoperative hypotension when using HPI monitoring in patients having elective major non-cardiac surgery.

## 1. Introduction

Intraoperative hypotension—i.e., low arterial blood pressure during surgery—is common in patients having non-cardiac surgery with general anesthesia [1]. Intraoperative hypotension is associated with postoperative acute myocardial and kidney injury [2,3,4,5,6] and postoperative mortality [2,7,8,9]. Intraoperative hypotension is a potentially modifiable risk factor for postoperative morbidity and mortality and should be avoided [10,11].

Avoiding intraoperative hypotension is a complex task for anesthesiologists [12]. Current blood pressure management is mainly reactive—with fluids and vasopressors being given to treat hypotension when it has already occurred. However, hypotension is a late clinical sign of hemodynamic instability and exhausted compensatory mechanisms. Predicting hypotension before it becomes clinically apparent, therefore, may allow preemptive treatment and help avoid hypotension [13]. Using artificial intelligence to predict hypotension from hemodynamic data is an innovative and intriguing approach.

The Acumen^TM^ Hypotension Prediction Index (HPI) software (Edwards Lifesciences; Irvine, CA, USA) was developed using artificial intelligence—specifically machine learning—and predicts hypotension from blood pressure waveform features [14]. The HPI is a unitless number ranging from 0 to 100. It indicates the likelihood that a patient will develop hypotension, defined as a mean arterial pressure (MAP) of less than 65 mmHg for at least one minute. The higher the HPI, the higher the likelihood that hypotension will occur shortly. HPI values over 85 trigger acoustic and visual alarms, a pop-up window, and a “secondary screen”. The secondary screen displays advanced hemodynamic variables to support clinicians to diagnose the cause of impending hypotension and initiate causal treatment.

We aimed to investigate whether using HPI monitoring helps avoid intraoperative hypotension. We built up a European, multicenter, prospective, observational registry to describe the incidence, duration, severity, and causes of intraoperative hypotension when using HPI monitoring in patients having elective major non-cardiac surgery.

## 2. Materials and Methods

EU-HYPROTECT is a European, multicenter, prospective, observational registry in patients having elective major non-cardiac surgery. The registry was conducted in accordance with the recommendations guiding physicians in biomedical research involving human subjects adopted by the 18th World Medical Assembly, Helsinki, Finland, 1964 and later versions, and in accordance with the European Medical Device Regulation (Regulation (EU) 2017/745 of 5 April 2017), and ISO 14155:2020. Ethics committee approval was obtained for each site prior to patient documentation. Patients provided written informed consent before study inclusion (unless waived by the local ethics committee due to the observational nature of the study). The study was registered at ClinicalTrials.gov (NCT04972266) on 22 July 2021 (Table 1). This article adheres to the Standard Protocol Items: Recommendations for Interventional Trials (SPIRIT) statement [15]. The first patient was enrolled in this registry on 27th September 2021.

### 2.1. Participating Sites

We collected data from 12 medical centers in 5 European countries (France, Germany, Italy, Spain, and United Kingdom). All participating sites had experience performing clinical research with patient-centered outcomes, agreed to the feasibility of recruitment, had electronic medical records containing procedural perioperative data of surgical patients, and verified the ability to recruit patients. Written informed consent was obtained from patients by the research team.

Patients in the registry were monitored using an advanced hemodynamic monitoring system, the Acumen^TM^ IQ sensor (Edwards Lifesciences) and the Hemosphere platform (Edwards Lifesciences), which calculates and continuously displays the HPI. Because using the HPI requires training and education, the registry was performed in medical centers that routinely use HPI monitoring with the Acumen^TM^ IQ sensor and the Hemosphere platform.

### 2.2. Patients

We planned to include at least 700 evaluable patients in the registry. We included consenting adults (≥18 years) who were scheduled for elective major non-cardiac surgery under general anesthesia that was expected to last at least 120 min and in whom arterial catheter placement and HPI monitoring was planned for clinical indications. We did not include patients having emergency surgery, nephrectomy, and liver or kidney transplantation; patients with atrial fibrillation and/or sepsis (according to current Sepsis-3 definition); patients with American Society of Anesthesiology physical status classification V or VI; patients who were not able to understand the nature, significance, and scope of the investigation; pregnant women; patients without signed informed consent/data protection statement; and patients participating in interventional trials.

### 2.3. Objectives and Outcomes

Once all the data will be collated, the major objectives will be to quantify and characterize intraoperative hypotension (defined as a mean arterial pressure (MAP) < 65 mmHg) when using HPI monitoring in patients having elective major non-cardiac surgery. This threshold was chosen because evidence from observational research suggests that the population harm threshold for postoperative organ injury is 60–70 mmHg for MAP [16] and the HPI software is trained to predict a MAP of 65 mmHg for at least one minute. Specifically, we will describe the time-weighted average (TWA) MAP < 65 mmHg (unit: mmHg)—that is the area under a MAP of 65 mmHg (unit: mmHg × minutes) divided by the total monitoring time (i.e., usually the total duration of surgery) (unit: minutes) [17,18]. We will also describe the TWA MAP < 60 mmHg and <55 mmHg; the area under a MAP of 65 mmHg, 60 mmHg, and 55 mmHg; the number of episodes of a MAP < 65 mmHg, <60 mmHg, and <55 mmHg; the proportion of patients with at least one episode (1 min or more) of a MAP < 65 mmHg, <60 mmHg, and <55 mmHg; and the absolute maximum decrease below a MAP of 65 mmHg, 60 mmHg, and 55 mmHg. We will also assess the primary causes of intraoperative hypotension by analyzing advanced hemodynamic variables during hypotensive episodes (including cardiac output, stroke volume, heart rate, systemic vascular resistance, pulse pressure variation, stroke volume variation, dP/dt_max_, and dynamic arterial elastance). On an exploratory basis, we will describe the incidence of (1) postoperative acute myocardial injury within 3 days after surgery, (2) postoperative acute kidney injury within 3 and 7 days after surgery, (3) death within 30 days after surgery, 4) hospital re-admission within 30 days after surgery, and (5) a composite outcome of non-fatal cardiac arrest and death within 30 days after surgery. For this purpose, we follow patients for 30 days after surgery (via a phone call if the patient leaves hospital earlier than 30 days after surgery).

Acute myocardial injury was defined as an increase in high-sensitivity troponin concentration within the first three postoperative days according to the definition of “myocardial injury and infarction associated with non-cardiac procedures” set forth in the Fourth Universal Definition of Myocardial Infarction (2018) [19]. We considered high-sensitivity troponin T or I values (whichever was used in each center) when measured per routine care before surgery (baseline) and on postoperative days 1, 2, or 3.

Acute kidney injury was defined based on “Kidney Disease: Improving Global Outcomes Clinical Practice Guideline for Acute Kidney Injury” [20,21] as (a) an increase in serum creatinine concentration of ≥0.3 mg/dL within any 48-h period within the first 7 postoperative days, (b) an increase in serum creatinine of ≥50% from baseline within the first 7 postoperative days, or (c) the need for renal replacement therapy within the first 7 postoperative days. We considered serum creatinine values when measured per routine care before surgery (baseline) and on postoperative days 1 to 7. We only considered the creatine and renal replacement criteria (i.e., excluding the urine output criterium) in accordance with current recommendations [20,21] because urine output is usually not reliably recorded in patients after surgery.

Non-fatal cardiac arrest was defined as successful resuscitation from ventricular fibrillation, ventricular tachycardia, asystole, or pulseless electrical activity requiring cardiopulmonary resuscitation, pharmacological therapy, or cardiac defibrillation.

Further, we will describe the intensive care unit length of stay and hospital length of stay. It will be documented whether an institutional hemodynamic protocol was available.

We will compare the incidence, duration, and severity of intraoperative hypotension observed in patients in the present registry with those reported in previous observational and interventional studies using similar metrics quantifying intraoperative hypotension and reporting postoperative complications (“historical data” from studies published in international peer-reviewed journals).

We will also compare the incidence, duration, and severity of intraoperative hypotension observed in patients in the EU-HYPROTECT registry with those observed in patients in whom HPI monitoring was not used. Hemodynamic data from such patients are available to Edwards Lifesciences because they are being shared under the umbrella of different joint projects between Edwards Lifesciences and hospitals using Edwards Lifesciences monitoring (data share agreements are in place with all institutions sharing patient data).

### 2.4. Data Collection and Management

Personal patient data remain at the participating centers. Pseudonymized clinical data (according to the data collection schedule set out in Table 2) were captured by an electronic case report form which can be reached via a secure website. To enter the data, investigators used a confidential and personalized login and password. The electronic case report form was designed to allow automatic and manual checks for plausibility and completeness. Each active site was visited, and all patients were monitored according to a predefined monitoring plan by the Contract Research Organization of the study (IPPMed; Institute for Pharmacology and Preventive Medicine GmbH, Cloppenburg, Germany or a respective representative). Per routine, monitoring visits confirmed that informed consent was obtained and checked that values recorded in the electronic case report form matched source documents. Data were also statistically audited using the method of Carlisle [22] and similar approaches. Pseudonymized data are accessible to the participating center, the principal investigator, and IPPMed or its authorized representatives. Pseudonymized data will be stored in the database for 10 years unless legal requirements demand longer storage. Adverse event data were sent to the competent national authorities and ethics committees involved. Protocol modifications were reported to ethics committees, investigators, and ClinicalTrials.gov. Clinical data from the registry will be published or presented using pseudonymized data only.

### 2.5. Statistics

The sample size was estimated based on published data. A randomized controlled trial reported a TWA MAP < 65 mmHg of 0.44 mmHg (25% and 75% percentiles, 0.23 and 0.72 mmHg) without and 0.10 mmHg (25% and 75% percentiles, 0.01 and 0.43 mmHg) with HPI monitoring [18]. Considering the wide interquartile range of TWA MAP < 65 mmHg, we planned to include at least 700 evaluable patients in this registry.

Statistical analysis will be performed for the total registry population, as well as for subgroups. We will perform descriptive analyses to describe and quantify intraoperative hypotension, patient characteristics, and perioperative data. For continuous variables, means with standard deviations will be presented for normally distributed data and medians with 25th and 75th percentiles for non-normally distributed data. For categorical variables, numbers and percentages will be shown. Patients with missing data will be excluded from the analysis. All statistical analyses will be performed using IBM SPSS Statistics (IBM, Armonk, NY, USA) or R Core Team (Version 4.2.1; R Core Team/R Foundation for Statistical Computing, Vienna, Austria; https://www.R-project.org/; date accessed: 19 September 2022).

### 2.6. Data Monitoring

There were no monitoring visits prior to site and patient enrollment. Physicians and registry personnel were required to make themselves familiar with the registry protocol, eCRF, requirements, and procedures.

According to a predefined monitoring plan, a fixed percentage of sites were randomly selected and monitored by IPPMed. In these centers, source data verification was performed according to a pre-specified monitoring plan. If findings from data management will raise doubts with regard to data quality at a specific site, additional monitoring visits may be performed (risk-based monitoring).

## 3. Discussion

With the data from this registry, we will describe the incidence, duration, severity, and causes of intraoperative hypotension when using HPI monitoring in patients having elective major non-cardiac surgery. The HPI software uses machine learning to predict hypotension from changes in blood pressure waveform features.

Intraoperative hypotension has been identified as a common and potentially modifiable risk factor for postoperative organ injury in patients having non-cardiac surgery [10,11]. Anesthesiologists should, therefore, consider avoiding intraoperative hypotension as a mainstay of intraoperative hemodynamic management. However, avoiding intraoperative hypotension is challenging [12] because hypotension can have numerous causes, including vasodilation due to anesthetic drugs or systemic inflammation, hypovolemia due to bleeding, or low cardiac output due to impaired cardiac contractility. The pathophysiological rationale behind using machine learning-based predictive monitoring is that predicting hypotension before it becomes clinically apparent may allow preemptive or timely treatment and help avoid hypotension [13]. This pathophysiological rationale is reasonable because hypotension is preceded by subtle hemodynamic changes that are imperceptible for the human eye and current monitoring methods [23,24]. Real-time hemodynamic monitoring combined with machine learning may identify these subtle hemodynamic changes preceding hypotension [11,23,24,25].

Machine learning, in general, is the study of computer algorithms that automatically learn and improve with experience using large data sets and thus allowing the identification and analysis of features from a vast amount of data. Over the past decade, the development and application of machine learning algorithms across a range of industries—including healthcare—has grown substantially. Using machine learning to predict hemodynamic instability is an intriguing approach that has the potential to initiate a paradigm shift in hemodynamic monitoring and management—from reactive to predictive monitoring and proactive management.

The HPI algorithm was developed by Hatib and co-workers [14] for the real-time prediction of hypotension. The HPI algorithm was modeled and cross-validated in 1334 surgical and critically ill patients using 545,959 min of arterial blood pressure waveform recordings and 25,461 episodes of hypotension [14]. It was externally validated in 204 surgical patients [14]. The HPI predicts hypotension—in the current version defined as a MAP < 65 mmHg for at least 1 min—from 23 individual arterial blood pressure waveform features (that were identified from 3022 individual and 2,606,147 combinatorial waveform features) [14].

Validation studies in surgical patients suggest that the HPI may predict hypotension 15 min before the event with a sensitivity and specificity of more than 80% [14,26]. HPI-guided blood pressure management reduced intraoperative hypotension compared to routine care in small preliminary trials in patients having hip arthroplasty [27] and general non-cardiac surgery—but not in a trial randomizing 212 non-cardiac surgery patients to HPI-guided or routine blood pressure management [28].

We included consenting adults having elective major non-cardiac surgery with general anesthesia in whom arterial catheter placement and HPI monitoring was planned for clinical indications independent of the study. We used duration of surgery (at least 120 min) to select patients having major surgery who are at particular risk of developing hypotension. Although the HPI algorithm also works on blood pressure waveforms recorded non-invasively using a finger-cuff device [29,30], we only included patients in whom blood pressure is monitored invasively using an arterial catheter because intraarterial blood pressure monitoring remains the clinical reference method [31]. Patients having emergency surgery were not included because these patients would not have enough time to provide informed consent. Patients with atrial fibrillation were not included because the HPI algorithm is not validated in patients with atrial fibrillation. Sepsis causes marked alterations in cardiovascular dynamics—including alterations in vasomotor tone and blood pressure regulation. We, therefore, also excluded septic patients from the registry.

To quantify intraoperative hypotension, we will report the TWA MAP < 65 mmHg—i.e., the area under a MAP of 65 mmHg divided by the total duration of surgery [17,18]. TWA MAP reflects the combination of the duration and severity of hypotension—and thus is a measure of hypotension that is clinically meaningful. In a preliminary randomized trial, the median TWA MAP < 65 mmHg was 0.44 mmHg without HPI monitoring and 0.10 mmHg with HPI monitoring [18]. However, TWA MAP is not routinely used in clinical practice and, thus, clinicians will not be familiar with this measure. We will, therefore, also describe measures of hypotension that are frequently used in clinical practice—including the number of episodes below certain MAP thresholds and the proportion of patients with at least one episode (1 min or more) below certain MAP thresholds.

Intraoperative hypotension can have different causes [10,32]. A detailed understanding of underlying causes could allow treating intraoperative hypotension causally with specific interventions. We will, therefore, also assess causes of intraoperative hypotension by analyzing advanced hemodynamic variables during hypotensive episodes (including cardiac output, stroke volume, heart rate, systemic vascular resistance, and pulse pressure variation).

The goal of any intraoperative hemodynamic monitoring and management strategy is to improve postoperative patient-centered outcomes. The registry will not have sufficient power to investigate the effect of HPI monitoring on patient-centered outcomes. Nevertheless, we will describe the incidence of postoperative acute myocardial and kidney injury, postoperative death, hospital re-admission, and a composite outcome of non-fatal cardiac arrest and death on an exploratory basis. We considered stroke as potential further outcome as it is a major posteroperative complication. However, overt (not covered) postoperative strokes occur rarely compared to other major postoperative complications. Additionally, there are conflicting data on a potential association between intraoperative hypotension and postoperative stroke in non-cardiac surgery patients (with many studies indicating that there is no strong association [2,33,34]). Finally, we discussed whether delirium is a potentially valuable outcome. Although it is common, the initial diagnosis of delirium mostly relies on suspicion, and often will go undetected or is misdiagnosed. Furthermore, a thorough clinical evaluation is considered the gold standard for its diagnosis, and there is no biomarker with high sensitivity and specificity. However, investigating postoperative delirium would require assessing patients several times a day during the first postoperative days. Although European Guidelines [35] advocate the post-operative assessment of delirium, it was decided not to document it in the current study.

While a randomized trial would have been desirable, available data on intraoperative hypotension when HPI monitoring is used in a multicenter setting in Europe are scarce. As such, there was a lack of data for a sample size calculation and further observational data were required. The results of this registry study will inform the design of future randomized trials investigating whether HPI-guided blood pressure management improves patient-centered outcomes in patients having surgery compared to routine blood pressure management. Specifically, the registry will provide event rate estimates necessary for robust sample size estimations for randomized trials.

## 4. Trial Status

The protocol version is 1.4 with an effective date of 19 May 2022. Patient recruitment began in September 2021 and was completed in May 2022. The 30-day follow-up was completed in June 2022.

## Figures and Tables

**Table 1 jcm-11-05585-t001:** Trial registration data.

Data Category	Information
Primary registry and trial identifying number	ClinicalTrials.gov, identifier: NCT04972266
Date of registration in primary registry	22 July 2021
Secondary identifying numbers	n.a.
Source(s) of monetary or material support	Edwards Lifesciences SA, Route de l’Etraz 70, 1260 Nyon, Switzerland
Primary sponsor	Edwards Lifesciences SA, Route de l’Etraz 70, 1260 Nyon, Switzerland
Secondary sponsor(s)	n.a.
Contact for public queries	IPPMed—Institute for Pharmacology and Preventive Medicine GmbH, Cloppenburg, Germany email: claudia.lueske@ippmed.de, daniel.greinert@ippmed.de
Contact for scientific queries	Prof. Dr. Bernd Saugel Department of Anesthesiology Center of Anesthesiology and Intensive Care Medicine University Medical Center Hamburg-Eppendorf, Hamburg, Germany email: b.saugel@uke.de
Public title	The EU-HYPROTECT Registry
Scientific title	Acumen^TM^ Hypotension Prediction Index software to prevent intraoperative hypotension during major non-cardiac surgery (EU-HYPROTECT): study protocol for a European multicenter prospective observational registry
Countries of recruitment	France, Germany, Italy, Spain, United Kingdom
Health condition(s) or problem(s) studied	Intraoperative hypotension, postoperative complications
Intervention(s)	na
Key inclusion and exclusion criteria	Inclusion criteria: consenting adults (≥18 years) who were scheduled for elective non-cardiac surgery under general anesthesia that was expected to last at least 120 min and in whom arterial catheter placement was planned for clinical indications independent of the study and in whom hypotension prediction index monitoring was planned. Exclusion criteria: patients having emergency surgery, nephrectomy, and liver or kidney transplantation; patients with atrial fibrillation and/or sepsis (according to current Sepsis-3 definition); patients with American Society of Anesthesiology physical status classification V or VI; patients who were not able to understand the nature, significance, and scope of the investigation; pregnant women; patients without signed informed consent/data protection statement; and patients participating in interventional trials.
Study type	Multicenter prospective observational registry
Date of first enrolment	27 September 2021
Target sample size	700 patients evaluable
Recruitment status	Recruitment complete
Key outcome(s)	To describe the time-weighted average (TWA) mean arterial pressure (MAP) < 65 mmHg when using Hypotension Prediction Index monitoring in patients having elective major non-cardiac surgery. TWA MAP < 60 mmHg and < 55 mmHg; the area under a MAP of 65 mmHg, 60 mmHg, and 55 mmHg Number of episodes of a MAP < 65 mmHg, < 60 mmHg, and < 55 mmHg Proportion of patients with at least one episode (1 min or more) of a MAP < 65 mmHg, < 60 mmHg, and < 55 mmHg Absolute maximum decrease below a MAP of 65 mmHg, 60 mmHg, and 55 mmHg Assess causes of intraoperative hypotension.

MAP: mean arterial pressure, n.a.: not applicable, TWA: time-weighted average.

**Table 2 jcm-11-05585-t002:** Data collection schedule.

	Screening Visit	Baseline Visit	Surgery Visit	Postoperative Data	Registry Exit
Inclusion/exclusion criteria	X				
ASA classification	X				
Signed informed consent ^1^		X			
Demographics		X			
Comorbidities		X			
Medications		X			
Lab values		X		X	
Type of surgery			X		
Vital signs			X		
Procedural details, anesthesia & surgery			X		
Safety parameter/complications			X	X	
Length of hospital stay				X	
Registry exit					X

^1^ Consent needed to be given prior to the procedure. ASA, American Society of Anesthesiologists physical status classification; X, data were collected at the specific timepoints.

## Data Availability

Aggregated data may be available from the Sponsor upon reasonable request. Data will be accessible by the sponsor, the Contract Research Organization, and the steering committee. Local data will be accessible for each center.

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
