# Peer review of "Hypotension Prediction Index Software to Prevent Intraoperative Hypotension during Major Non-Cardiac Surgery: Protocol for a European Multicenter Prospective Observational Registry (EU-HYPROTECT)"

_jcm, 2022, doi:10.3390/jcm11195585_

Round 1
Reviewer 1 Report
Manucript jcm-1835460
Dear Editor, Dear Authors,
Thank you very much for giving me an opportunity to review this manuscript. It is well written with adequate flow. Also, the subject under investigation is very important to the medical field and as such should be investigated with due detail. Monitoring based on machine learning may be the future of intraoperative detection of hypotension before it becomes clinically visible and thus may save health and lives of many patients by allowing the avoidance of hypotension and immediate treatment when it occurs.
I read the manuscript with great interest, yet some elements raise my major concern.
1. My first major concert is the list of authors.
Please explain how it is possible that among 14 authors of this manuscript there is only one woman. Modern medicine must be inclusive to be serious. The list of authors for this manuscript is a common example that women in research are excluded and marginalized. I fully understand that once the manuscript is written a change of authors is not possible, yet as a reviewer for JCM and a part of the editorial board I must raise this as a major issue and draw you attention to this problem.
Ref:
Kamerlin SCL, Wittung-Stafshede P. Female Faculty: Why So Few and Why Care? Chemistry. 2020 Jul 8;26(38):8319-8323. doi: 10.1002/chem.202002522. Epub 2020 Jun 25. PMID: 32583921.
Ref:
Squazzoni F, Bravo G, Grimaldo F, García-Costa D, Farjam M, Mehmani B. Gender gap in journal submissions and peer review during the first wave of the COVID-19 pandemic. A study on 2329 Elsevier journals. PLoS One. 2021 Oct 20;16(10):e0257919. doi: 10.1371/journal.pone.0257919. PMID: 34669713; PMCID: PMC8528305.
2. My second major concern is the definition intraoperative hypotension
In lines 162-164 the authors write that “The major objectives are to quantify and characterize intraoperative hypotension (defined as a mean arterial pressure (MAP) <65 mmHg) when using HPI monitoring in patients having elective major non-cardiac surgery.” This sentence has no reference, and the definition seems to have been chosen arbitrary. According to a recent review by Weinberg et al. IOH should be defined using the absolute values stated in the POQI statement i.e., MAP < 60–70 mmHg or SBP < 100 mmHg
Ref.
Weinberg, L., Li, S.Y., Louis, M. et al. Reported definitions of intraoperative hypotension in adults undergoing non-cardiac surgery under general anaesthesia: a review. BMC Anesthesiol 22, 69 (2022). https://doi.org/10.1186/s12871-022-01605-9
3. My third major concern is the list of observed outcomes that does not include perioperative stroke occurring within the 30 days of surgery. Please explain why this is not a part of the follow-up and analysis. The introduction misses an important point – it has been shown that there is an association between intraoperative hypotension and perioperative ischemic stroke. This should be highlighted in this section and mentioned in the abstract, included in the outcomes and final analysis, but if this is not possible – the lack of this information should be mentioned in the limitations.
Ref. Bijker JB, Persoon S, Peelen LM, Moons KG, Kalkman CJ, Kappelle LJ, van Klei WA. Intraoperative hypotension and perioperative ischemic stroke after general surgery: a nested case-control study. Anesthesiology. 2012 Mar;116(3):658-64. doi: 10.1097/ALN.0b013e3182472320. PMID: 22277949.
Ref: NeuroVISION Investigators. Perioperative covert stroke in patients undergoing non-cardiac surgery (NeuroVISION): a prospective cohort study. Lancet. 2019 Sep 21;394(10203):1022-1029. doi: 10.1016/S0140-6736(19)31795-7. Epub 2019 Aug 15. PMID: 31422895.
4. Moreover, postoperative delirium (POD) is also one of the outcomes that may be associated with intraoperative hypotension. When doing a prospective trial, one would expect that collecting information regarding POD and its association with IOH is part of the protocol.
Ref.
Wachtendorf LJ, Azimaraghi O, Santer P, Linhardt FC, Blank M, Suleiman A, Ahn C, Low YH, Teja B, Kendale SM, Schaefer MS, Houle TT, Pollard RJ, Subramaniam B, Eikermann M, Wongtangman K. Association Between Intraoperative Arterial Hypotension and Postoperative Delirium After Noncardiac Surgery: A Retrospective Multicenter Cohort Study. Anesth Analg. 2022 Apr 1;134(4):822-833. doi: 10.1213/ANE.0000000000005739. PMID: 34517389.
5. Lines 189-190 cite the KDIGO criteria, but the original reference is not cited. According to the KDIGO criteria, both the creatinine and the urine output criteria should be used to identify all cases of AKI. The authors chose to use only the creatinine criteria, please explain why.
With best regards
Reviewer 2 Report
If a comparison between episodes of hypotension between AI monitored and non AI monitored patients is planned, an appropriate study design would be an RCT.
Please check extensively for grammar and specially the tense used in the manuscript. All events which have happened before this version of protocol have to be in past tense and the plan should be future/ future perfect tense.
This happens to be version 1.4 of the protocol, last modified in May 2022. Why was there a need to modify the protocol after recruitment started? what were the modifications made? Will this affect the data collection of those who were enrolled before the modifications?
Round 2
Reviewer 1 Report
Dear Authors,
Thank you for providing the answers. The manuscript has been improved, yet some issues remain.
Ad 1.1 Please provide a full list of researchers contributing to the EU-HYPROTECT study as a supplement for this manuscript.
Ad. 1.2 No further issues.
Ad. 1.3 No further issues at this point. However the idea of publishing the protocol of a study in advance is to do it before the beginning of data collection, not after, to include the reviewers' suggestions and adapt the protocol accordingly. It seems that the publication of the protocol of this study is rather delayed if the authors state that "the data collection is largely complete".
Ad. 1.4 I understand that POD cannot be added as an endpoint to this study, yet I do not agree with the provided explanation. POD monitoring is recommended by the ESAIC until the 5th postoperative day and as such should be included as an outcome in all prospective studies. There are tools for POD detection and writing that an outcome recommended by ESAIC to be evaluated in all patients is "too complex to be documented reliably" is not right. Please rephrase.
Ref. Aldecoa C, Bettelli G, Bilotta F, Sanders RD, Audisio R, Borozdina A, Cherubini A, Jones C, Kehlet H, MacLullich A, Radtke F, Riese F, Slooter AJ, Veyckemans F, Kramer S, Neuner B, Weiss B, Spies CD. European Society of Anaesthesiology evidence-based and consensus-based guideline on postoperative delirium. Eur J Anaesthesiol. 2017 Apr;34(4):192-214. doi: 10.1097/EJA.0000000000000594. Erratum in: Eur J Anaesthesiol. 2018 Sep;35(9):718-719. PMID: 28187050.
Ad. 1.5 No further issues.
With best regards
Author Response
Reviewer 1
Dear Authors,
Thank you for providing the answers. The manuscript has been improved, yet some issues remain.
Comment 1.1: Please provide a full list of researchers contributing to the EU-HYPROTECT study as a supplement for this manuscript.
Response 1.1: There is an investigator list in the appendix of the manuscript that we amended for further research staff involved as suggested. Among them there are many female researchers (physicians as well as nurses).
Comment 1.2: No further issues.
Response 1.2: Thank you
Comment 1.3: No further issues at this point. However, the idea of publishing the protocol of a study in advance is to do it before the beginning of data collection, not after, to include the reviewers' suggestions and adapt the protocol accordingly. It seems that the publication of the protocol of this study is rather delayed if the authors state that "the data collection is largely complete".
Response 1.3: We appreciate your position. The study protocol was written by the steering committee members and approved by the ethics committees responsible for each of the centers. The protocol is also listed on clinicaltrials.gov and the entry mirrors the latest released version of the study protocol.
We feel that the final study protocol should be published for transparency purposes and as a reference to upcoming results publications.
Comment 1.4: I understand that POD cannot be added as an endpoint to this study, yet I do not agree with the provided explanation. POD monitoring is recommended by the ESAIC until the 5th postoperative day and as such should be included as an outcome in all prospective studies. There are tools for POD detection and writing that an outcome recommended by ESAIC to be evaluated in all patients is "too complex to be documented reliably" is not right. Please rephrase.
Ref. Aldecoa C, Bettelli G, Bilotta F, Sanders RD, Audisio R, Borozdina A, Cherubini A, Jones C, Kehlet H, MacLullich A, Radtke F, Riese F, Slooter AJ, Veyckemans F, Kramer S, Neuner B, Weiss B, Spies CD. European Society of Anaesthesiology evidence-based and consensus-based guideline on postoperative delirium. Eur J Anaesthesiol. 2017 Apr;34(4):192-214. doi: 10.1097/EJA.0000000000000594. Erratum in: Eur J Anaesthesiol. 2018 Sep;35(9):718-719. PMID: 28187050.
Response 1.4: We rephrased the explanation in the manuscript to match with your suggestions.
Change 1.4: “Finally, we discussed whether delirium is a potentially valuable outcome. Although it is common, the initial diagnosis mostly relies on suspicion, and often will go undetected or is misdiagnosed. Furthermore, a thorough clinical evaluation is considered the gold standard for its diagnosis, and there is no biomarker with high sensitivity and specificity. However, investigating postoperative delirium would require assessing patients several times a day during the first postoperative days. Although European Guidelines [Aldecoa C et al. Eur J Anaesthesiol 2017; 34: 192-214] advocate the post-operative assessment of delirium, it was decided not to document it in the current study.”
Comment 1.5: No further issues.
Response 1.5: Thank you
With best regards
